# p16 deficiency attenuates intervertebral disc degeneration by adjusting oxidative stress and nucleus pulposus cell cycle

Hui Che[1,2†], Jie Li[3†], You Li[1], Cheng Ma[1], Huan Liu[1,4], Jingyi Qin[5], Jianghui Dong[6,7], Zhen Zhang[6], Cory J Xian[7], Dengshun Miao[8], Liping Wang[6,7]*, Yongxin Ren[1]*

[1]Department of Orthopaedics, The First Affiliated Hospital of Nanjing Medical University, Nanjing, China; [2]University Medical Center, Albert-Ludwigs-University, Freiburg, Germany; [3]Department of Orthopaedics, Xuzhou Central Hospital, Xuzhou Clinical College of Nanjing Medical University, The Affiliated Xuzhou Hospital of Southeast University, Xuzhou, China; [4]The Affiliated Huai'an No.1 People's Hospital of Nanjing Medical University, Huai'an, China; [5]Suzhou Institute of Nano-Tech and Nano-Bionics, Chinese Academy of Sciences, Suzhou, China; [6]Department of Hand Surgery, Department of Plastic Reconstructive Surgery, Ningbo, China; [7]School of Pharmacy and Medical Sciences and UniSA Cancer Research Institute, University of South Australia, Adelaide, Australia; [8]State Key Laboratory of Reproductive Medicine, The Research Center for Bone and Stem Cells, Department of Anatomy, Histology and Embryology, Nanjing Medical University, Nanjing, China

*For correspondence:
liping.wang@mymail.unisa.edu.au
(LW);
renyongxinjsph@163.com (YR)

†These authors contributed equally to this work

Competing interests: The authors declare that no competing interests exist.

**Abstract** The cell cycle regulator p16 is known as a biomarker and an effector of aging. However, its function in intervertebral disc degeneration (IVDD) is unclear. In this study, p16 expression levels were found to be positively correlated with the severity of human IVDD. In a mouse tail suspension (TS)-induced IVDD model, lumbar intervertebral disc height index and matrix protein expression levels were reduced significantly were largely rescued by p16 deletion. In TS mouse discs, reactive oxygen species levels, proportions of senescent cells, and the senescence-associated secretory phenotype (SASP) were all increased, cell cycling was delayed, and expression was downregulated for Sirt1, superoxide dismutase 1/2, cyclin-dependent kinases 4/6, phosphorylated retinoblastoma protein, and transcription factor E2F1/2. However, these effects were rescued by p16 deletion. Our results demonstrate that p16 plays an important role in IVDD pathogenesis and that its deletion attenuates IVDD by promoting cell cycle and inhibiting SASP, cell senescence, and oxidative stress.

## Introduction

Intervertebral disc degeneration (IVDD) refers to the physiological and pathological process of natural degeneration and aging of the intervertebral disc, which is the basis of various clinical spinal diseases (*Silagi et al., 2018*). IVDD usually results in vertebral instability, disc herniation, and spinal canal stenosis, which are commonly accompanied by low back pain with or without symptoms of nerve root or spinal cord compression. These lead to tremendous societal and economic burdens worldwide. It has been estimated that, at some point during their lifetime, 80–90% of the world's population suffer clinical low back pain, which is positively correlated with disc degeneration in approximately 23% of these people (*Smith et al., 2011*; *Walker, 2000*).

**eLife digest** Neck and shoulder pain, lower back pain and leg numbness are conditions that many people will encounter as years go by. This is because intervertebral discs, the padding structures that fit between the bones in the spine, degenerate with age: their cells enter a 'senescent', inactive state, and stop multiplying.

A protein known as p16, an important regulator of cell growth and division, is known to accumulate in senescent cells. In fact, in mouse fat tissue, muscles or eyes, removing the cells that contain high levels of p16 delays aging-associated disorders. However, it was still unknown whether deactivating the gene that codes p16 in senescent cells could delay disc degeneration.

Here, Che, Li et al. discovered that p16 is highly present in the senescent cells of severely degenerated human intervertebral discs. The cells in the nucleus pulposus, the jelly-like and most critical tissue in the intervertebral discs, were extracted and grown in the lab under conditions that replicate the early stages of damage to the spine. Drugs and genetic manipulations were then used to decrease the amount of p16 in these cells.

The experiments showed that reducing the levels of p16 results in the senescent cells multiplying more and showing fewer signs of damage and aging. In addition, the discs of mice in which the gene that codes for p16 had been deleted were less prone to degeneration compared to 'normal' mice in similar conditions.

Overall, the work by Che, Li et al. shows that inhibiting p16 in disc cells delays the aging process and reduces the degeneration of intervertebral discs. These findings may one day be applicable to people with intervertebral disc diseases who, for example, could potentially benefit from a gene therapy targeting the cells which produce p16.

The human intervertebral disc is a non-vascular tissue, and its annulus fibrosus (AF) and inner layer nucleus pulposus (NP) rely mainly on the penetration of the end plate to provide nutrition. In this chronically high osmotic pressure, low pH, hypoxic and low-nutrition environment, the cells are less active. This is one of the reasons for the poor self-healing ability of the disc's structure and function after tissue damage, and thus, intervertebral discs degenerate more easily than other tissues in the body (*Feng et al., 2016*). Among the many factors that cause intervertebral disc degeneration is the accumulation of senescent disc cells (most of which are NP cells), which has provided a novel insight into IVDD pathogenesis. Senescent NP cells generate only a small number of new cells; therefore, the number of functional cells decreases gradually. Moreover, senescent NP cells may change the disc microenvironment, creating a senescence-associated phenotype in which proinflammatory factors are overexpressed, extracellular matrix (ECM) is decreased, and growth factors and chemokines are downregulated (*Le Maitre et al., 2007*; *Markova et al., 2013*; *van Deursen, 2014*). However, the molecular mechanisms that underpin cell senescence in IVDD are unclear.

Cell senescence is regulated by various molecular signaling pathways. One of the canonical molecules involved in cell senescence is p16 (p16INK4a), which is encoded by the *CDKN2A* gene and belongs to the cell cycle regulatory pathway (*Serrano, 1997*). Senescent cells, most of which seem to express p16 (*Childs et al., 2017*), accumulate with aging and are conducive to tissue dysfunction. The clearance of p16-positive senescent cells in adipose tissue, skeletal muscle and the eye has been suggested to delay aging-associated disorders in mice (*Baker et al., 2011*). Specifically, the systemic clearance of p16-positive senescent cells and conditional *Cdkn2a* gene deletion have been shown to mitigate age-associated IVDD in mice, mostly by suppressing the senescence-associated secretory phenotype (SASP), improving matrix homeostasis, and reducing apoptosis (*Novais et al., 2019*; *Patil et al., 2019*). However, we do not yet know how p16 drives disc cell senescence and whether other factors are present in the progression of IVDD, especially in human discs.

Increasing levels of reactive oxygen species (ROS), another main feature of aging, are involved in a number of age-related pathologies. Senescence can occur under prolonged oxidative states; and thus, ROS is seen as an important mediator of the progression of cellular senescence (*Colavitti and Finkel, 2005*). Pathological ROS levels have been implicated in the induction of senescence-like phenotypes similar to that of p16-induced senescence. An increasing number of studies have shown that

p16 might play a role in oxidative stress-associated senescence (*Gonçalves et al., 2016*; *Mas-Bargues et al., 2017*). Nonetheless, whether p16 contributes to intervertebral disc aging by increasing ROS is unclear. The present study aimed to highlight the influence of p16 on disc degeneration, mainly focusing on oxidative stress and human NP cell proliferation, and verified this effect in mice that have homozygous deletion of *Cdkn2a*.

## Results

### p16 was upregulated in the NP of degenerated human intervertebral discs

To explore the role of p16 in IVDD, p16 expression was first verified in the NP tissues of patients with various degrees of disc degeneration, as examined by histological staining (Pfirrmann grades 2–5, *Figure 1—figure supplement 1*). H and E staining showed a substantially disordered tissue texture in samples with a high Pfirrmann grade, in which hypertrophic and vacuole-like cells and multinuclear giant cells were present at the end of the sample (*Figure 1A*). Masson and Safranin O staining showed smaller amounts of proteoglycans (PGs) and increased levels of fibrosis in NP tissues that had a high Pfirrmann grade (*Figure 1A*). These results confirm that severely degenerated NP tissue is correlated with a high Pfirrmann grade. In addition, immunohistochemistry (IHC) and western blotting (WB) revealed that degenerated discs with a higher Pfirrmann grade expressed a higher level of p16 than those with a lower Pfirrmann grade (*Figure 1B,C,D;* with p16 expression levels and their corresponding Pffirmann grades being shown in *Figure 1—figure supplement 2*). These results confirm that p16 accumulates in NP tissues as IVDD progresses.

### p16 regulated NP cell proliferation and senescence under IL-1β stimulation by mediating oxidative stress and the cell cycle

To uncover how p16 participates in IVDD progression, NP cells with mild degeneration were isolated from Pfirrmann grade 2 tissues and cultured in vitro. IL-1β was used to induce NP cell degeneration. Immunofluorescent (IF) and SA-β-gal staining clearly illustrated that IL-1β greatly increased the percentage of senescent NP cells and p16 protein expression when compared with control levels (*Figure 2A,D,E*). In addition, the effect of altering p16 expression levels on NP cell degeneration and proliferation was investigated. p16 expression was down- and upregulated by siRNA or plasmid transfection, respectively. The transfection efficiencies of targeted siRNAs and plasmids, compared with those of null siRNA and empty plasmid, respectively, are presented in *Figure 2—figure supplement 1*. p16 expression decreased after siRNA-mediated knockdown, which decreased the proportion of NP cells that demonstrated a senescent phenotype. By contrast, p16 overexpression caused a marked opposite effect (*Figure 2B,H*). In addition to the degree of aging, proliferative ability is another indicator of cellular degeneration. Cell counting kit-8 (CCK-8) analyses confirmed that NP cell proliferation was reduced by p16 overexpression and promoted by p16 knockdown when compared with the levels seen after treatment with IL-1β treatment alone (*Figure 2C*).

To determine the potential mechanism by which p16 modulates NP cell physiological behavior, flow cytometry was used to analyze ROS levels and the cell cycle. ROS levels were obviously higher when p16 expression was increased (*Figure 2F,G*). As p16 expression gradually increased from the p16+IL-1β group to the IL-1β group and the control group, the cells presented cell-cycle arrest in the $G_0/G_1$ phase. Interestingly, considerably more NP cells progressed through $G_0/G_1$ to S phase following p16 downregulation (*Figure 2F,I*). These results demonstrate that p16 might regulate senescence by mediating oxidative stress and promoting proliferation by accelerating the movement of cells through the $G_1/S$ checkpoint.

### Rapamycin inhibited p16 expression and promoted NP cell proliferation by reducing oxidative stress and mediating the cell cycle

Recent studies have reported that rapamycin prevents IVDD by inhibiting cell senescence via the mTOR signaling pathway (*Choi et al., 2016*; *Ito et al., 2017*). Therefore, the application of rapamycin was used to inhibit cell senescence in order to explore the interplay between this cellular event and p16 in IVDD prevention. Rapamycin antagonized the effect of IL-1β, decreasing p16 expression

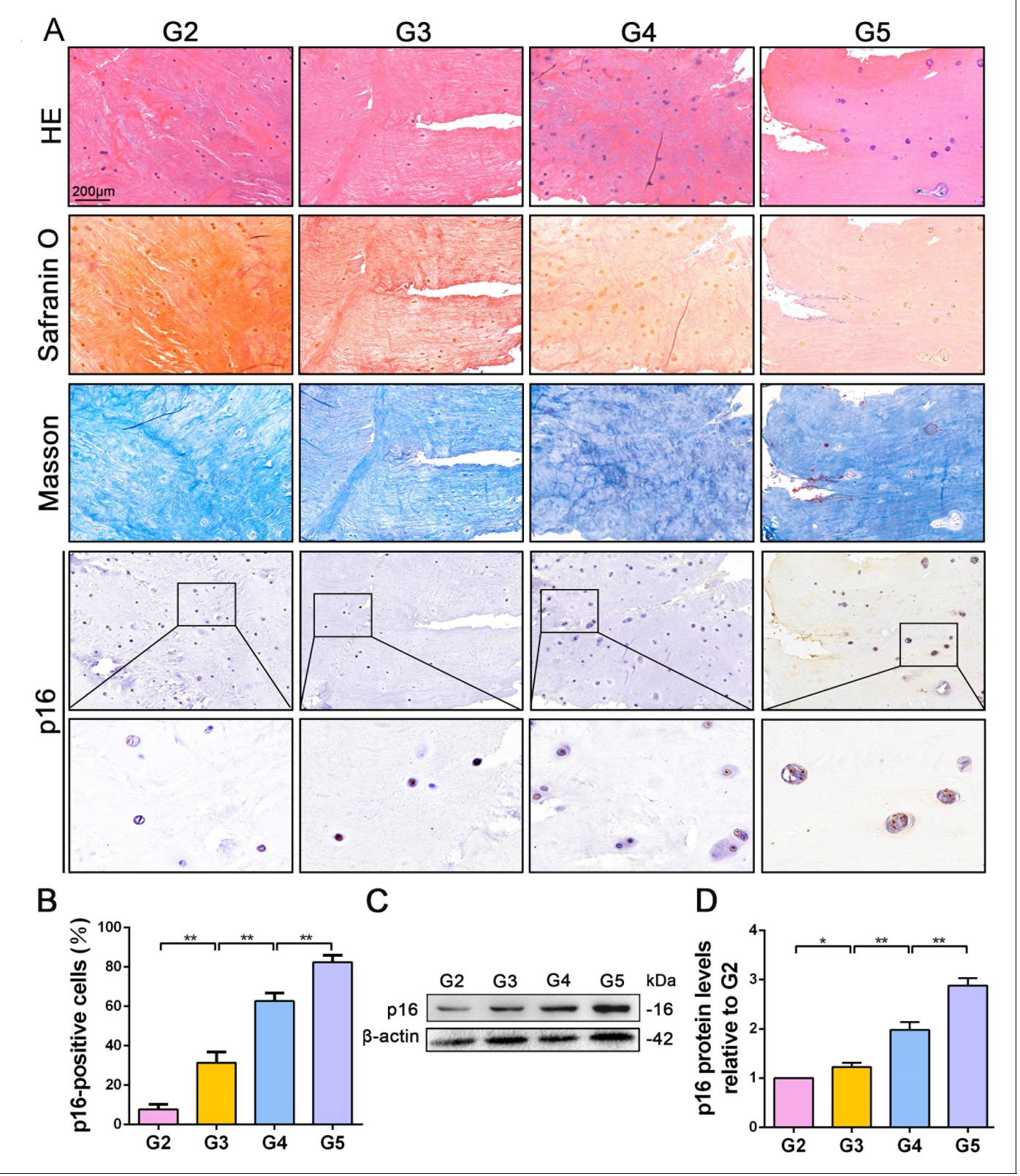

**Figure 1.** p16 expression in NP cells from human interverbal discs with different degrees of degeneration (G2–G5 groups according to Pfirrmann grade). (**A**) Representative images of H and E staining showing cell/tissue general morphology. Safranin O staining with collagen and NP cells appearing orange and fibers blue/violet. Masson staining with collagen and NP cells appearing blue and fibers red; and immunohistochemical staining

*Figure 1 continued on next page*

*Figure 1 continued*

for p16. (B) Quantification of p16-positive cells (%). p16 protein levels were assessed by (C) western blotting and (D) measured by densitometric analyses and expressed as folds relative to grade 2 (G2) NP samples. Data are presented as mean ± SD (n = 3); *p<0.05; **p<0.01.

The online version of this article includes the following source data and figure supplement(s) for figure 1:

**Source data 1.** Source data and related summary statistics for *Figure 1A and C*.

**Figure supplement 1.** Representative magnetic resonance imaging (MRI) scans of patients with different categories of disc degeneration according to Pfirrmann grade.

**Figure supplement 2.** Pffirmann grade of the degenerated disc from the human specimens correlates individually with p16 expression. ***p<0.001.

**Figure supplement 2—source data 1.** Related summary statistics.

---

and the percentage of senescent NP cells (*Figure 3A,B,C,E,F*). Flow cytometry analyses revealed significantly lower ROS levels in the IL-1β+rapamycin group than in the IL-1β-alone group, and showed that rapamycin treatment reduced the ratio of cells in the $G_0/G_1$ phase; meanwhile, the ratio of cells in S phase was higher after rapamycin treatment (*Figure 3D,G,H*). CCK-8 assays also showed a higher level of proliferation in the IL-1β+rapamycin group when compared with the IL-1β alone group (*Figure 3I*). These changes parallel the above described impact of p16 siRNA. These findings demonstrate that rapamycin can prevent NP cell degeneration by reducing ROS levels and mediating the cell cycle, which might be associated with p16 inhibition.

## p16 deletion partly postponed mouse IVDD

To further assess whether p16 deletion plays a positive role in IVDD prevention in vivo, the *Cdkn2a* gene knock out (p16 KO) mice and the tail suspension (TS) method were used to establish a mouse IVDD model. After 4 weeks of TS, muscles around the spine were congested with varying degrees of injury (*Figure 4—figure supplement 1B*). Based on the morphological and histological changes among different groups, disc height index (DHI) analyses showed that mouse disc heights were decreased by TS but were maintained in p16 KO mice when compared with WT mice (*Figure 4A,C*). Furthermore, micro-magnetic resonance imaging (MRI) demonstrated that TS reduced water content in the disc and that p16 deletion significantly protected against this effect (*Figure 4H*, *Figure 4—figure supplements 2*, *3*). After TS, disc heights decreased and more vesicular cells appeared, and the discs in p16 KO mice exhibited obviously higher glycosaminoglycan (GAG) levels with or without TS than those in WT mice (*Figure 4B*).

Inflammation is a vital part of the disc degeneration process. To examine the effects of systemic p16 knockout, the levels of inflammatory factors in the NP tissues of mice were analyzed. There was a clear difference in inflammation between p16 KO and WT mice, as p16 deletion reduced the expression of TNF-α, IL-1β, and IL-6 (*Figure 4G*). These results were confirmed when RNA expression was assessed (*Figure 4F*). Furthermore, because NF-κB-p65 has a vital function in regulating inflammatory responses and is activated by various stimuli, including stress, we conducted analyses on NF-κB-p65 expression levels. Western blot analyses showed higher p65 levels in the TS group than in the control group. After p16 gene deletion, p65 expression decreased significantly (*Figure 4D,E*).

Matrix metallopeptidases (MMPs) can degrade all types of ECM proteins, decreasing the aggrecan and collagen II content of the tissue. Thus, we evaluated MMP3, MMP9, MMP10, and MMP13 mRNA and protein expression levels. The levels of these MMPs mostly increased after TS, and p16 deletion partly reversed these changes (*Figure 4D,E*). Treatment effects on the expression levels of the typical components of ECM, aggrecan, collagen I, collagen II, and collagen X, were measured by western blot or qRT-PCR. Aggrecan and collagen II, which are protective ECM components, were slightly degraded in p16 KO mice compared to WT mice; whereas collagens I and X, which are harmful ECM components, were expressed at low levels in p16 KO mice (*Figure 4D,E*). In summary, these results suggest that p16 deletion partly postpones IVDD in mice as assessed in terms of changes in disc height, water content, inflammation, and ECM components.

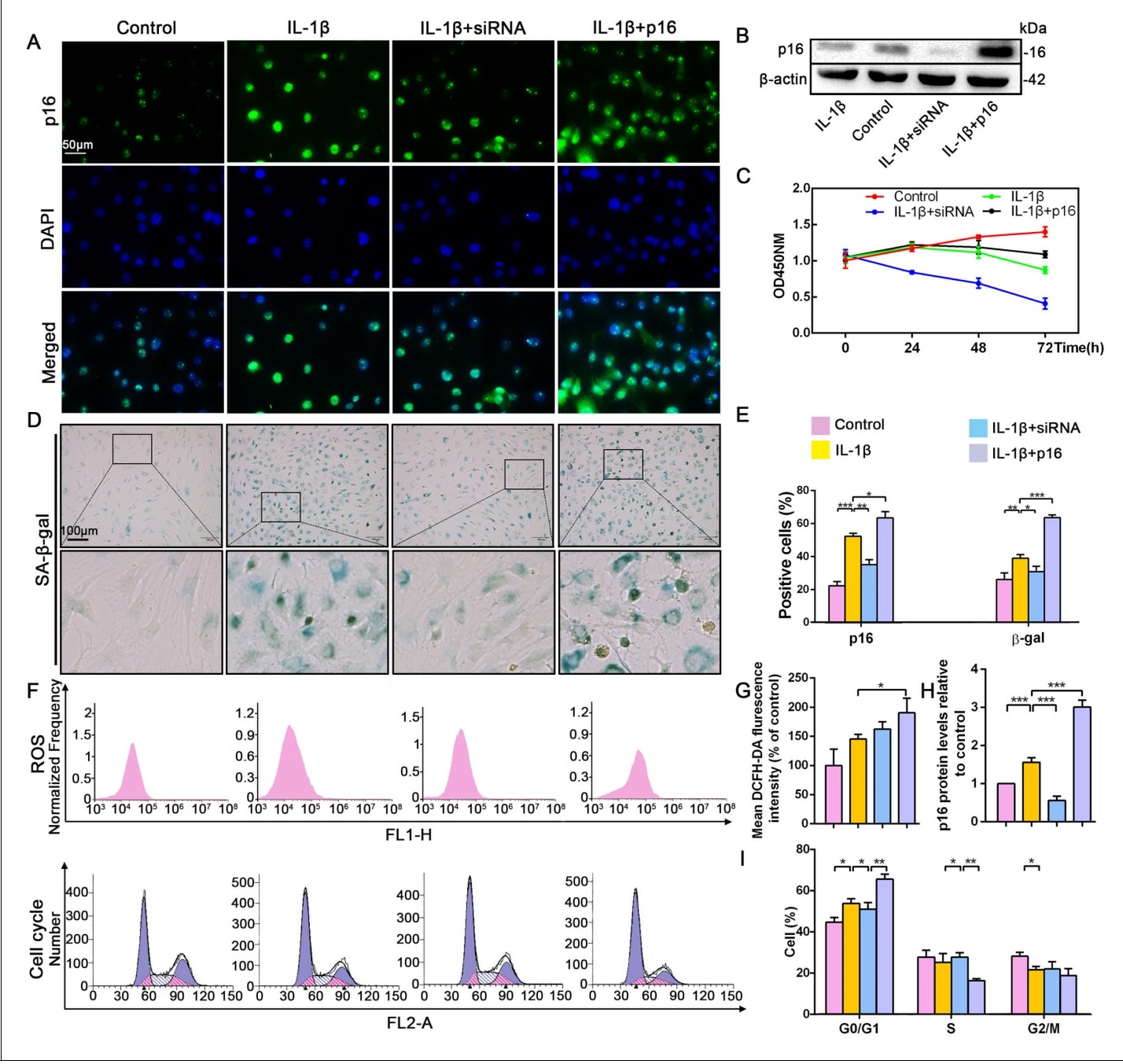

**Figure 2.** Effect of p16 on senescence, reactive oxygen species (ROS) levels and NP cell proliferation upon IL-1β stimulation (10 ng/mL). Human NP cells were grouped as follows: normal cultured cells (control), IL-1β treated cells (IL-1β), p16-siRNA-transfected cells treated with IL-1β (IL-1β+siRNA), and p16 plasmid-transfected cells treated with IL-1β (IL-1β+p16). (A) Representative immunofluorescent micrographs stained for p16. (B) p16 protein levels as assessed by western blotting. (C) Cell proliferation as assessed by CCK-8 assays. (D) SA-β-gal staining. (E) Total p16-positive and β-gal-positive cells (%). (F) ROS levels and the cell-cycle distribution of freshly collected human NP cells as determined by flow cytometry. (G) Quantitation of ROS levels. (H) p16 level measured by densitometric analysis and expressed relative to the control. (I) Cell-cycle distribution. Data are presented as mean ± SD (n = 3); *p<0.05; **p<0.01; ***p<0.001.

The online version of this article includes the following source data and figure supplement(s) for figure 2:

**Source data 1.** Source data and related summary statistics for *Figure 2B, C, D and F*.

**Figure supplement 1.** Efficiency of transfection with p16 siRNA and the p16 plasmid compared with that with null siRNA and empty plasmid.

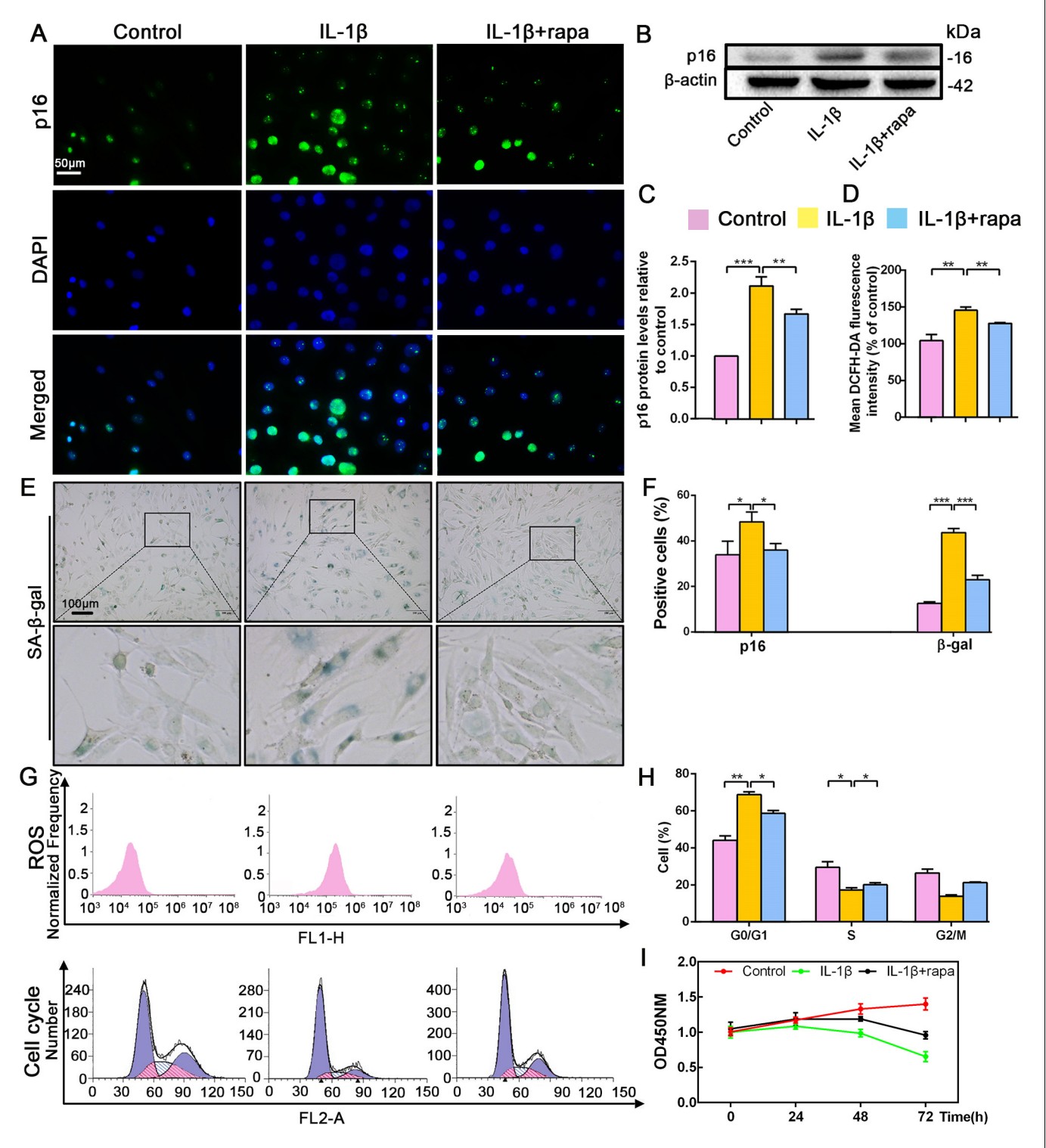

**Figure 3.** Effect of rapamycin (50 nM) on senescence, reactive oxygen species (ROS) levels and NP cell proliferation upon IL-1β stimulation (10 ng/mL). Human NP cells were grouped as follows: normal cultured cells (control), IL-1β treated cells (IL-1β), and rapamycin-stimulated cells treated with IL-1β (IL-1β+rapa). (A) Representative immunofluorescent micrographs stained for p16. p16 protein levels as (B) assessed by western blotting and (C) measured by densitometric analysis, with results expressed relative to the control. (D) Quantitation of ROS levels. (E) SA-β-gal staining. (F) Total p16-positive and β-gal-positive cells (%). (G) ROS levels and the cell-cycle distribution of freshly collected human NP cells as determined by flow cytometry. (H) Cell-cycle distribution. (I) Cell proliferation as assessed by CCK-8 assays. Data are presented as mean ± SD (n = 3). *p<0.05; **p<0.01; ***p<0.001.
The online version of this article includes the following source data for figure 3:

**Source data 1.** Source data and related summary statistics *Figure 3B, E, G and I*.

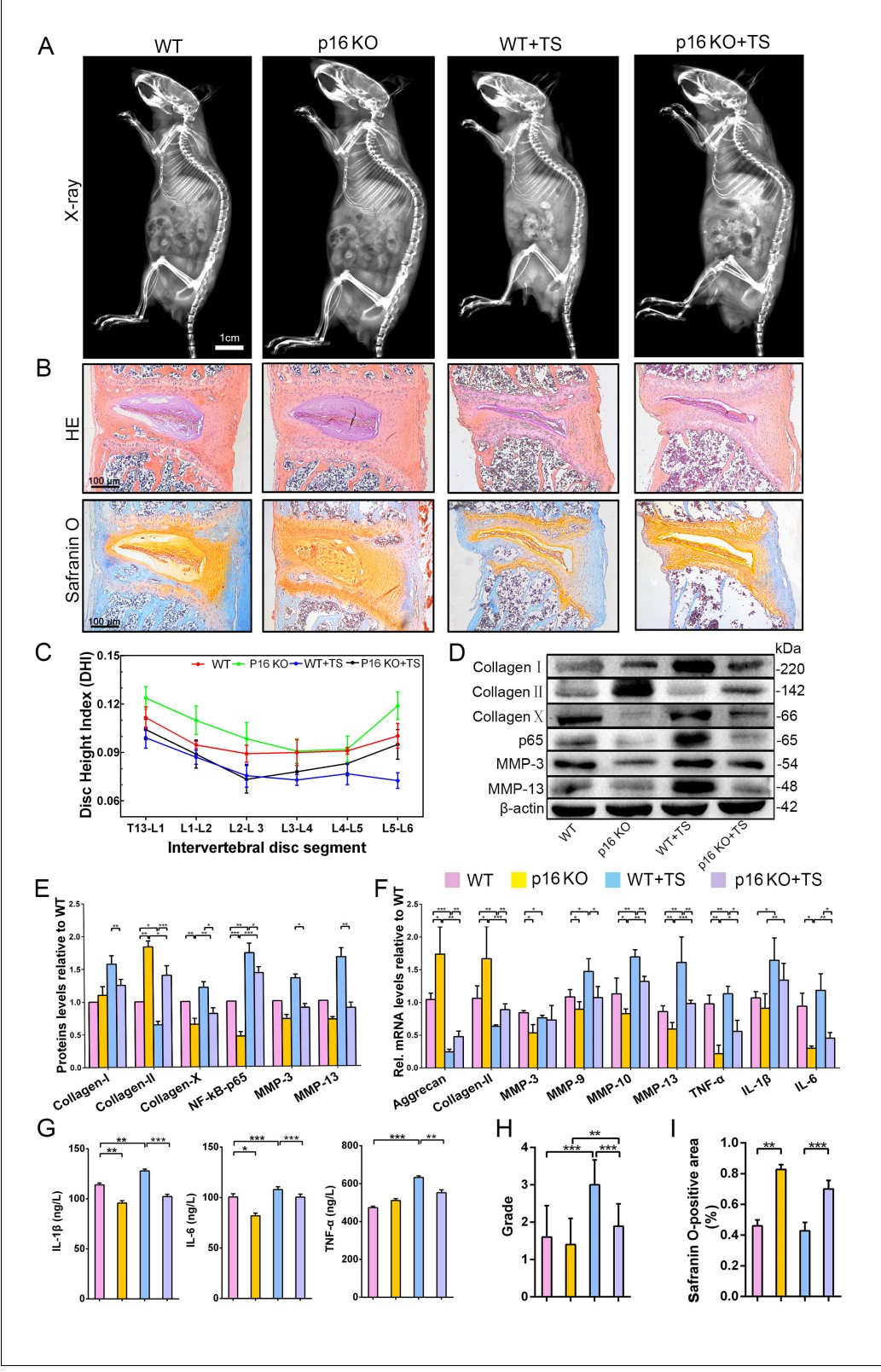

**Figure 4.** p16 deletion delayed mouse intervertebral disc degeneration (IVDD). WT and p16 KO mice were fed on the ground or with tail suspension (TS). (**A**) Radiographs of overall mouse length. (**B**) After H and E staining and Safranin O staining, collagen and NP cells are orange, and fibers are blue. (**C**) The intervertebral disc height index as calculated on the basis of lumbar vertebrae. (**D**) Associated protein levels as assessed by western blotting and (**E**) as measured by densitometric analysis, with results expressed relative to those in WT mice. (**F**) Target mRNA expression assessed by RT-PCR

*Figure 4 continued on next page*

*Figure 4 continued*

relative to GAPDH expression. (**G**) IL-1β, IL-6 and TNF-α levels in disc tissues as determined by ELISA. (**H**) The modified Thompson classification as assessed on the basis of lumbar disc signals. (**I**) Safranin O-positive area (%). Data are presented as mean ± SD (n = 3); *p<0.05; **p<0.01; ***p<0.001.

The online version of this article includes the following source data and figure supplement(s) for figure 4:

**Source data 1.** Source data and related summary statistics for *Figure 4B, C, D, G and H*.
**Figure supplement 1.** Establishment of TS-induced mouse IVDD model.
**Figure supplement 2.** Representative micro-MRI of mouse intervertebral discs.
**Figure supplement 3.** Pffirmann grade of degenerated disc from the mice specimens.
**Figure supplement 3—source data 1.** Related summary statistics.
**Figure supplement 4.** The intervertebral disc height index (DHI) was calculated by averaging the measurements obtained from the (**A**) posterior, (**B**) middle, and (**C**) anterior portions of the intervertebral disc and dividing these values by the average height of the adjacent (**D–I**) posterior, middle, and anterior portions of the vertebral body.

## p16 deletion exerted an antioxidant effect and affected progression from $G_1$ to S phase in vivo

To further determine the potential mechanism by which p16 functions in IVDD in vivo, multiple biological indicators were explored. Specifically, we assessed the degree of senescence, proliferative capacity, oxidative stress level, and the expression of cell-cycle proteins in p16 KO and WT mice with or without TS. IHC analyses of β-gal and western blot of p19 and p53 revealed that the discs of WT mice exhibited a more senescent phenotype than those of p16$^{-/-}$ mice in both the TS and control groups (*Figure 5A,C,D,F*). The proportions of PCNA- and Ki67-positive cells, the percentages of proliferative cells, and the IGF1 protein levels in the discs were substantially higher in p16 KO mice than in WT mice, even after TS (*Figure 5A*). Interestingly, the levels of vascular endothelial growth factor (VEGF), a microangiogenesis marker, were decreased in p16 KO mice, suggesting a protective function of p16 in disc degeneration (*Figure 5C,D*).

Because ROS levels decreased in human NP cells upon silencing p16, as described above, the antioxidant enzyme gene expression and ROS levels were determined in mouse disc tissues. SOD1, SOD2, GPX1, GPX3, and CAT mRNA expression increased upon p16 deletion, and p16 KO mice had lower total ROS levels than did WT mice, even after TS (*Figure 5B,G,I*). The mouse IVDD model also revealed the same effects on Sirt1, SOD1, and SOD2 protein levels (*Figure 5C,D*). The DNA injury marker 8-hydroxy-deoxyguanosine (8-OHdG) can be induced by oxidative stress. When compared to WT mice, the proportions of 8-OHdG-positive cells were greatly decreased in p16 KO mice, indicating that p16 deletion plays a protective role in the antioxidant process in the disc (*Figure 5A*).

To explore whether p16 affects proliferation by mediating the cell cycle, cell-cycle progression and cell-cycle-related proteins were analyzed by flow cytometry and western blotting. p16 KO mice showed increased progression of cells from $G_0/G_1$ into S phase compared with WT mice, with or without TS (*Figure 5B,H*). CDK4, CDK6, pRb, E2F1 and E2F2 protein expression levels were upregulated in p16 KO mice compared to WT mice. Conversely, the expression level of RB protein was downregulated in p16 KO mice (*Figure 5C,E*). These results demonstrate that p16 deletion can partially inhibit aging-related senescence by reducing disc oxidative stress injury and enhancing NP cell proliferation by promoting progression through the $G_1$/S checkpoint.

## NF-κB-p65 promoted p16 expression in human NP cells by activating the p16 promoter

Because expression of the transcription factor NF-κB-p65 differed between p16 KO and WT mice, it was hypothesized that NF-κB-p65 might control p16 protein levels. To confirm that NF-κB-p65 controls p16 at the transcriptional level, five putative NF-κB-p65 binding sites in the *CDKN2A* promoter region were identified and chromatin immunoprecipitation (ChIP) primers were designed using Primer Premier (*Supplementary file 1*). First, when we assessed whether NF-κB-p65 binds to the five putative promoter sequences, only two sites were verified by ChIP as efficient binding sites. One putative promoter sequence that was bound effectively is shown in *Figure 6A*, and another promoter sequence with no binding is shown in *Figure 6—figure supplement 1*. Using human genomic DNA as a template, the whole *CDKN2A* promoter

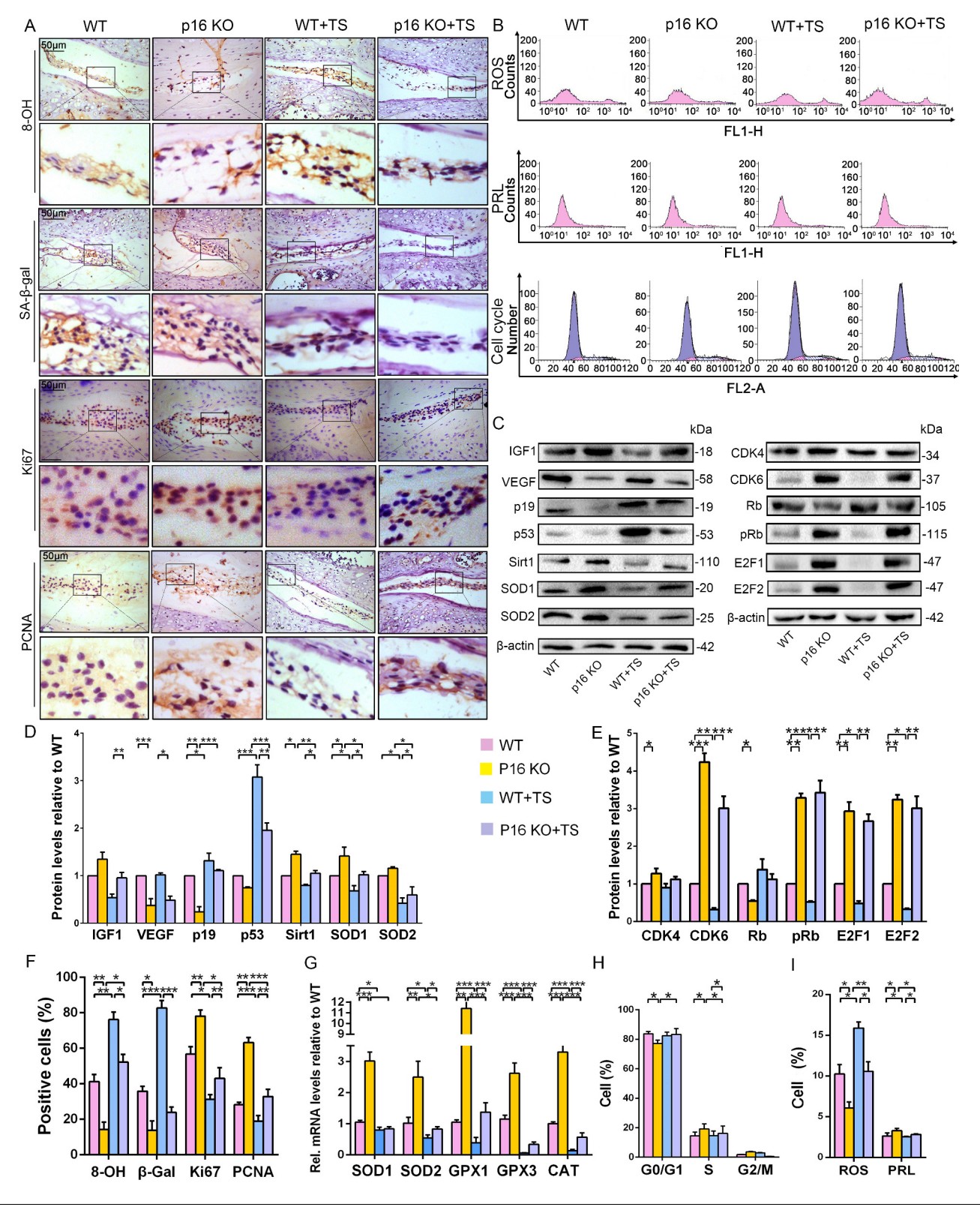

**Figure 5.** p16 deletion exerted an antioxidant effect and promoted mouse NP cell proliferation in vivo. WT and p16 KO mice were fed on the ground or with tail suspension (TS). (**A**) Representative micrographs of slices stained immunohistochemically for 8-hydroxy-2 deoxyguanosine (8-OHdG), senescence-associated β-galactosidase (SA-β-gal), Ki67 and proliferating cell nuclear antigen (PCNA). (**B**) Reactive oxygen species (ROS) levels, cell proliferation (PRL) and cell-cycle distribution in freshly collected mouse NP cells, as measured by flow cytometry. (**C**) Associated protein levels were

*Figure 5 continued on next page*

*Figure 5 continued*

assessed by western blotting and (D, E) measured by densitometric analysis with results expressed relative to those in WT mice. (F) Percentage of total immuno-positive cells (%). (G) Target mRNA expression as assessed by RT-PCR relative to GAPDH expression. (H) Cell-cycle distribution. (I) ROS and PRL (%) quantitation. Data are presented with mean ± SD (n = 3); *p<0.05, **p<0.01, ***p<0.001.

The online version of this article includes the following source data for figure 5:

**Source data 1.** Source data and related summary statistics for *Figure 5A, B and C*.

---

segment was amplified by PCR (lane 1 of *Figure 6A*). Clear DNA amplification was examined after immunoprecipitation without the irrelevant control IgG (lane 2 of *Figure 6A*), and with the anti-p65 antibody (lane 3 of *Figure 6A*).

Next, the WT and binding-site mutant *CDKN2A* promoter sequences were cloned into the pGL4.23-basic vector (producing pGL4.23-wt and pGL4.23-mut, respectively), and the resulting plasmids were transiently transfected into human NP cells (*Figure 6B*). Transfection with the empty plasmid (pGL4.23) without the *CDKN2A* promoter sequence and the Renilla expression plasmid (vector+pGL+pRL) or with the p65 plasmid, and pGL4.23 without the *CDKN2A* promoter sequence and the Renilla expression plasmid (p65+pGL+pRL) served as the negative controls. Luciferase activity was significantly higher in NP cells transfected with the p65 and pGL-wt plasmids than in those transfected with empty plasmid and the pGL-wt plasmid, indicating that p65 successfully activated the *CDKN2A* promoter. By contrast, luciferase activity was significantly lower in NP cells transfected with the p65 and pGL-mut plasmids than in those transfected with the p65 and pGL-wt plasmids (*Figure 6C*). The findings confirm that the *CDKN2A* promoter region with the predicted NF-κB-p65 binding sites is sufficient to promote transcription, providing a molecular mechanism that emphasizes p65-dependent p16 transcriptional activation in NP cells.

## Discussion

Although numerous studies have proven that p16 contributes to IVDD pathogenesis, few have focused on the role of p16 in humans. Here, an unbiased comparison of p16 expression in NP cells from IVDD patients with varying Pfirrmann scores showed that p16 expression is positively correlated with the degree of human disc degeneration. Along with the increase in the severity of human IVDD, NP cells showed decreased PG contents, increased fibrosis, and greater vacuolization; moreover, more multinucleated giant cells were observed, and p16 accumulated. The findings show that p16 plays a role in IVDD progression. Using NP cells harvested from patients with disc degeneration with a Pfirrmann score of 2 (mild disc degeneration), in-vitro studies further explored the function of p16 in the pathology of disc degeneration. The results suggest that p16 deficiency decreases oxidative stress and DNA damage in NP cells and contributes to NP cell proliferation, which protects against IVDD by promoting cell-cycle progression. In addition, for potential therapeutic exploration, these findings have provided theoretical and experimental evidence to support the potential use of rapamycin or the upstream approach targeting NF-κB-p65 to suppress p16 expression. Thus, the current investigation has not only demonstrated the different mechanisms of p16 in IVDD but also may provide theoretical evidence to inform the exploration of effective methods to downregulate p16 in order to reverse IVDD.

Although the presence of p16-positive cells indicates that an organism is in an inactive state (*Baker et al., 2011*), it is unclear whether the differential expression of p16 affects human disc degeneration. To simulate the microenvironment in disc degeneration, IL-1β was used to induce NP cell senescence. Multiple analyses revealed that p16 expression increased significantly as the degree of senescence increased in NP cells. However, the senescent phenotype of NP cells became less pronounced when p16 was silenced. By contrast, increased NP cell senescence was observed when p16 levels were upregulated by plasmid transfection. These results imply that p16 not only is produced by NP cell senescence but also accelerates NP cell senescence.

ROSs are mainly induced during cellular aging, but the mechanism by how p16 regulates ROS levels in NP cells is not yet clear. The ROS levels in different groups of NP cells expressing different levels of p16 showed that ROS levels increased along with p16 overexpression. This result suggests a strategy to reduce ROS in NP cells via p16 suppression.

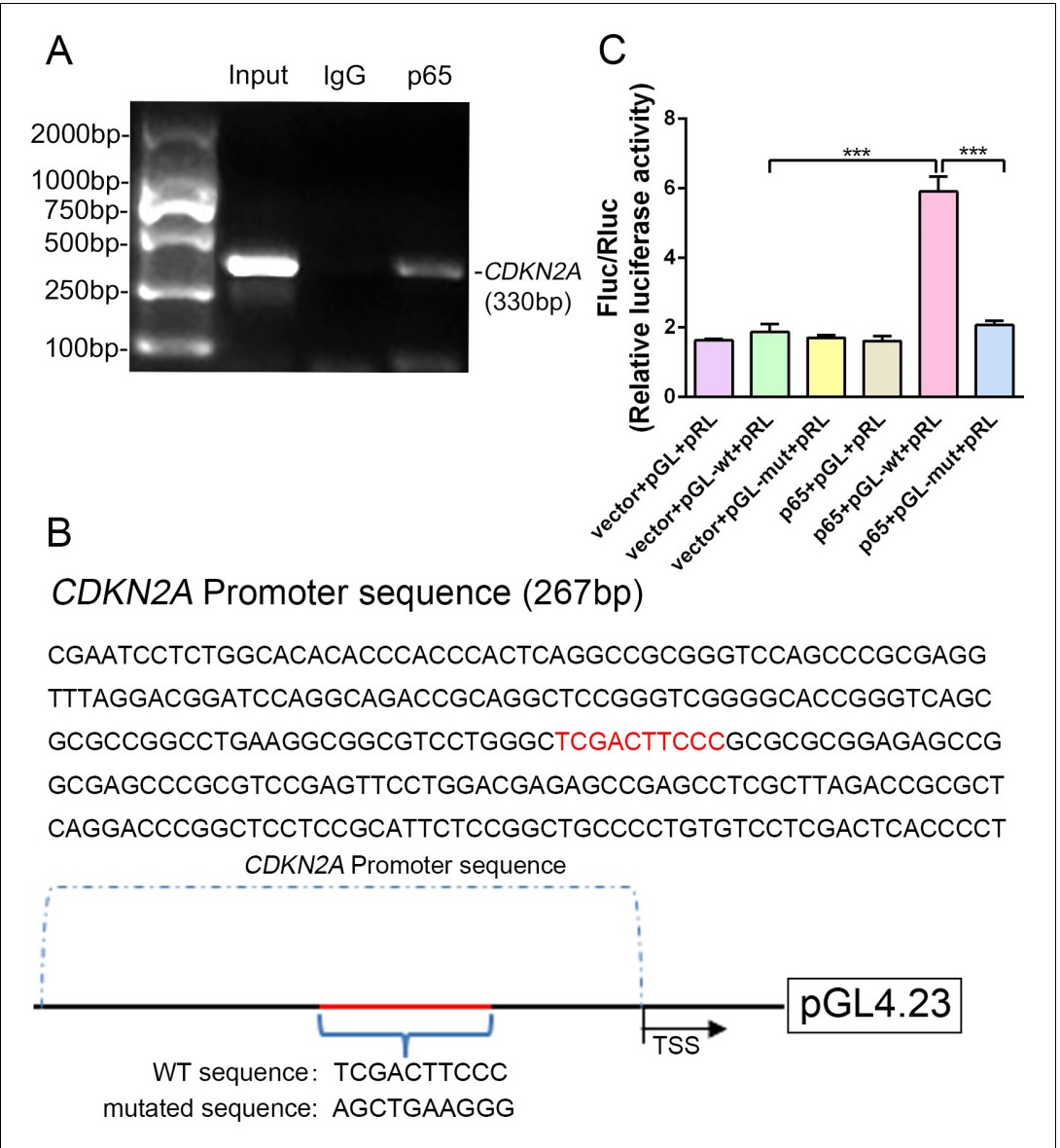

**Figure 6.** NF-κB-p65 bound the *CDKN2A* gene promoter and promoted p16 expression in human NP cells. (A) *CDKN2A* promoter sequences were recovered by PCR from p65 immunoprecipitates. (B) p65-like elements in the human *CDKN2A* promoter region and the mutated sequence are marked in red (upper panels). *Below*: structural schematic of the WT and mutant pGL4.23-p16 promoter reporter plasmids. (C) Luciferase activity driven by the *CDKN2A* promoter was more pronounced following NF-κB treatment. By contrast, luciferase activity that was not driven by the *CDKN2A* luciferase reporter decreased in the absence of NF-κB, and luciferase activity not driven by the mutant *CDKN2A* luciferase reporter decreased upon NF-κB treatment. Data are shown with mean ± SD (n = 3); ***p<0.001.

The online version of this article includes the following source data and figure supplement(s) for figure 6:

**Source data 1.** Source data and related summary statistics for *Figure 6C*.

**Figure supplement 1.** Another site in NF-κB-p65 that is predicted to bind the *CDKN2A* promoter.

**Figure supplement 1—source data 1.** Source data and related summary statistics for *Figure 6—figure supplement 1C*.

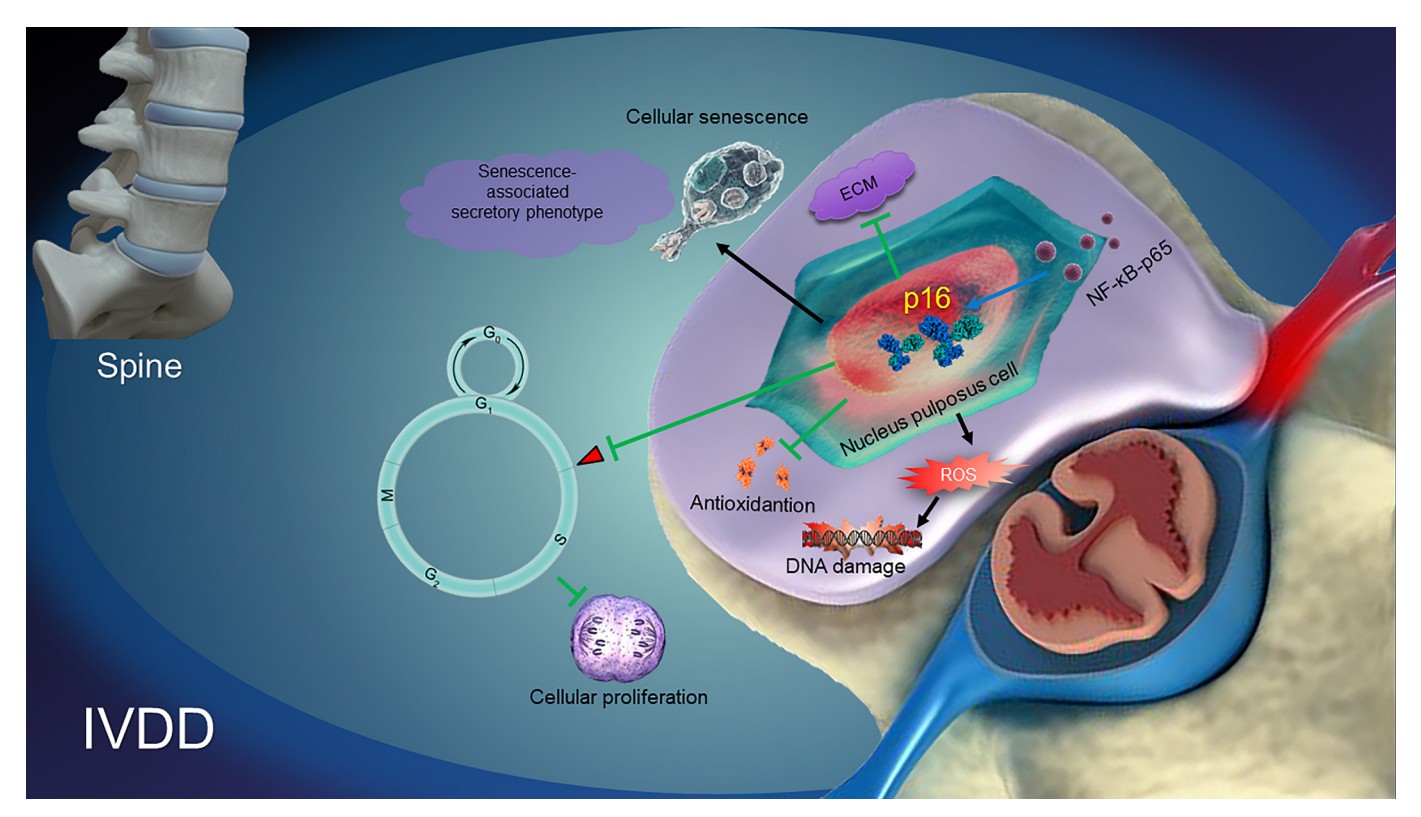

**Figure 7.** A proposed model for the mechanism of p16 in regulating intervertebral disc degeneration (IVDD). NF-κB-p65 activates p16 expression. p16 deficiency alleviates the reactive oxygen species (ROS) levels, senescence-associated secretory phenotype (SASP) and cellular senescence. Subsequently, p16 deficiency promotes the activity of cellular antioxidation, and the proliferation and stability of ECM, like aggrecan and collagen II. All of the pathways ultimately protect against the development of IVDD.

Senescence plays a basic role in regulating cell cycling by halting cell proliferation, and p16 expression was observed to be negatively correlated with human NP cell proliferation, an effect that was reversed by p16 downregulation. Previously, p16 was shown to be a cyclin-dependent kinase (CDK) inhibitor that is sufficient to inhibit cell proliferation and to induce aging features in mammals by restraining the cell cycle (*Boquoi et al., 2015*). The in-vitro model illustrated that p16 can inhibit the progression of NP cells from $G_1$ to S phase, which is an essential mechanism by which proliferation is regulated in human NP cells.

On the basis of the results described above, it is more feasible to suppress p16 expression using a specific drug rather than siRNA transfection. After all, for patients, drug therapy is more convenient and economical than gene therapy. Rapamycin was previously shown to have antiaging effects in multiple cells and organisms by modulating oxidative stress, nutrient sensing, and the cell cycle (*Richardson, 2013*; *Wang et al., 2017a*). Rapamycin was also shown to inhibit p16 expression to some extent (*Gidfar et al., 2017*). However, previous studies on the effect of rapamycin on disc degeneration focused on the role of this compound in autophagy (*Ito et al., 2017*; *Tu et al., 2018*). Therefore, rapamycin was applied to inhibit p16 and to explore its function in regulating ROS levels and the cell cycle. Rapamycin significantly decreased p16 expression and reversed the senescent phenotype of human NP cells. Furthermore, rapamycin decreased ROS levels in NP cells, which showed increased proliferation. Cell-cycle analyses indicated that rapamycin promoted the progression of NP cells from $G_1$ to S phase. Therefore, rapamycin may suppress ROS levels and promote NP cell proliferation, and these effects may be related to its ability to suppress p16. Taken together, p16 downregulation is likely to exert an antioxidant effect and promotes human NP cell proliferation, therefore playing a protective role in IVDD.

To verify these results in vivo, compound mutant mice with homozygous *Cdkn2a* deletion were used to establish an IVDD model involving TS. As a shock absorber for the spine, the basic role of the disc is a mechanical one containing load distribution, energy dissipation, and motion permit in daily activities. Mechanical factors have been proposed as one of the mechanisms necessary for accelerating the aging progression of both human and rodent discs via altered loading in several studies, so the TS mice model is an applicable mechanical representation of both tensile force on human discs and the aging process (*Hutton et al., 2002*). Simulation of weightlessness by TS changes flexion-extension, axial rotation, lateral bending and hydrostatic pressure in the disc and leads to destruction of the ECM destruction, an inflammatory response and a catabolic process that represents premature aging-related IVDD progression (*Földes et al., 1996*).

In the present study, p16 deletion protected against changes in disc heights and disc water contents in mice, which are the most intuitive indicators of IVDD in the clinic. Levels of aggrecan and collagen II, which protect NP cells in the ECM, were significantly increased in p16 KO mice with or without TS. By contrast, the protein levels of the fibrosis markers collagens I and X decreased after p16 deletion. To demonstrate whether p16 deletion affects inflammation in mouse discs, some inflammatory factors were further examined, and the results showed that p16 deletion reduced both the protein and the mRNA levels of the inflammatory factors. Encouragingly, the measurements of ROS levels and antioxidant activity showed that p16 deletion improved antioxidant activity in the discs and decreased ROS levels and DNA injury. Furthermore, examinations of the proliferation markers Ki67, PCNA, and IGF-1 indicated increased proliferative capacity in NP cells after p16 deletion. Interestingly, angiogenesis, which has been shown to increase the risk of IVDD (*Kwon et al., 2017*; *Zaidi et al., 2018*), was reduced in p16 KO mice compared with WT mice; this might be a novel direction for future research. Finally, the analyses of the cell cycle and cell cycle-related proteins confirmed that p16 suppression activates CDK4 and CDK6 to promote Rb protein phosphorylation, enhance E2F1/E2F2 activity, and promote the progression of NP cells from $G_1$ to S phase. Taken together, these findings demonstrate that ablation of p16 can relieve mouse disc degeneration by protecting the ECM, inhibiting fibrosis, reducing inflammation, decreasing ROS levels, and promoting proliferation by regulating the cell cycle.

As p16 plays an important role in IVDD, it is essential to understand the molecular mechanism of p16 activation and dysfunction. Previous studies have shown that Bmi-1 and 1,25-dihydroxyvitamin D inactivate p16 (*Chen et al., 2019*; *Taylor et al., 2015*; *Yamakoshi et al., 2015*). However, the molecular mechanism of p16 activation is still unclear. Bmi-1 is reported to active NF-κB signaling in glioma angiogenesis, cell migration and invasion (*Jiang et al., 2013*; *Sun et al., 2014*). By contrast, 1,25-dihydroxyvitamin D takes part in the suppression of inflammation and anticancer properties by blocking NF-kB activation (*Chen et al., 2011*; *Fekrmandi et al., 2015*). Intriguingly, a decreased NF-kB-p65 level was observed in the discs of p16 KO mice compared with those of the WT. Therefore, further work is necessary to explore the relation between NF-kB-p65 and p16 in disc tissue. NF-κB-p65 is well known for its role in regulating inflammation, immune response, cell division and apoptosis, and has been shown to participate in IVDD progression (*Wang et al., 2018*; *Wang et al., 2017b*). The present results support the observation that NF-κB-p65 is involved in p16 activation. Furthermore, by conducting analyses with the JASPAR database of transcription factor binding profiles, five putative NF-κB-p65 binding sites were identified in the *CDKN2A* promoter, and the ChIP assays have confirmed the activity of two of these putative binding sites. Finally, these two binding sites were tested in luciferase reporter gene assays, which showed that NF-κB-p65 bound at one binding site in the *CDKN2A* promoter, confirming that NF-κB-p65 is upstream of p16.

Taken together, from the data in the current study, a model illustrating the role and possible mechanisms of p16 in regulating IVDD can be proposed (*Figure 7*). NF-κB-p65 probably increases p16 expression by promoter activation. p16 deficiency regulates the antioxidative behaviors of NP cells, resulting in the suppression of ROS levels and the alleviation of both NP cellular senescence and the SASP. Subsequently, p16 deficiency promotes cellular proliferation and the production of ECM, such as collagen II and aggrecan. It can be speculated that molecular studies of the p16 in the development of IVDD may provide new solutions for preventing degenerative disc diseases.

# Materials and methods

## Key resources table

| Reagent type (species) or resource | Designation | Source or reference | Identifiers | Additional information |
|---|---|---|---|---|
| Transfected construct (human) | p16 plasmid | Invitrogen | Addgene plasmid # 10916; RRID: Addgene_10916 | Vector backbone: pcDNA3 |
| Transfected construct (human) | Empty plasmid | Invitrogen | Addgene plasmid # 45346; RRID: Addgene_45346 | Vector backbone: pcDNA3 |
| Transfected construct (human) | p16 siRNA | GenePharma; Lau et al., 2007 | A09004 | |
| Transfected construct (human) | Null siRNA | GenePharma | A06001 | |
| Transfected construct (human) | pGL4.23-p16-wt plasmids | Promoterbio Lab | | pGL4.23-basic luciferase vector |
| Transfected construct (human) | pGL4.23-p16-mut plasmids | Promoterbio Lab | | pGL4.23-basic luciferase vector |
| Biological sample (human) | Nucleus pulposus cells | This paper | | Freshly isolated from human nucleus pulposus |
| Antibody | Anti-p16 ARC antibody (rabbit monoclonal) | Abcam | Cat# ab51243, RRID: AB_2059963 | IF (1:100); WB (1:1000) |
| Antibody | Anti-GLB1/ beta-galactosidase antibody (rabbit polyclonal) | Abcam | Cat# ab203749 | IHC (1:200) |
| Antibody | Anti-8-hydroxy-2'-deoxyguanosine antibody (mouse monoclonal) | Abcam | Cat# ab48508, RRID: AB_867461 | IHC (1:200) |
| Antibody | Anti-Ki67 antibody (rabbit polyclonal) | Abcam | Cat# ab15580, RRID: AB_443209 | IHC (1:200) |
| Antibody | Anti-PCNA antibody (rabbit monoclonal) | Abcam | Cat# ab92552, RRID: AB_10561973 | IHC (1:500) |
| Antibody | Anti-collagen I antibody (rabbit polyclonal) | Abcam | Cat# ab34710, RRID: AB_731684 | WB (1:1000) |
| Antibody | Anti-collagen X (rabbit polyclonal) | Abcam | Cat# ab58632, RRID: AB_879742 | WB (1:300) |
| Antibody | Anti-collagen II antibody (rabbit polyclonal) | Abcam | Cat# ab34712, RRID: AB_731688) | WB (1:1000) |
| Antibody | Anti-SIRT1 antibody (mouse monoclonal) | Abcam | Cat# ab110304, RRID: AB_10864359 | WB (1:500) |
| Antibody | Anti-superoxide dismutase one antibody (rabbit polyclonal) | Abcam | Cat# ab13498, RRID: AB_300402 | WB (1:500) |
| Antibody | Anti-SOD2/MnSOD antibody (rabbit polyclonal) | Abcam | Cat# ab13533, RRID: AB_300434 | WB (1:1000) |
| Antibody | Anti-MMP3 antibody (rabbit monoclonal) | Abcam | Cat# ab52915, RRID: AB_881243 | WB (1:1000) |
| Antibody | Anti-MMP13 antibody (rabbit polyclonal) | Abcam | Cat# ab39012, RRID: AB_776416 | WB (1:3000) |
| Antibody | Anti-beta actin antibody (mouse monoclonal) | Abcam | Cat# ab8226, RRID: AB_306371 | WB (1:1000) |
| Antibody | Anti-IGF1 antibody (rabbit polyclonal) | Abcam | Cat# ab9572, RRID: AB_308724 | WB (1:500) |
| Antibody | Anti-VEGF 165A antibody (mouse monoclonal) | Abcam | Cat# ab69479, RRID: AB_1271452 | WB (1:1000) |

*Continued on next page*

*Continued*

| Reagent type (species) or resource | Designation | Source or reference | Identifiers | Additional information |
|---|---|---|---|---|
| Antibody | Anti-Cdk6 (rabbit polyclonal) | Abcam | Cat# ab131469, RRID: AB_11156738 | WB (1:1000) |
| Antibody | Anti-Cdk4 (rabbit monoclonal) | Abcam | ab199728 | WB (1:2000) |
| Antibody | Anti-p53 antibodies (mouse monoclonal) | Santa Cruz Biotechnology | Cat# sc-126, RRID: AB_628082 | WB (1:1000) |
| Antibody | Anti-p19 antibody (mouse monoclonal) | Santa Cruz Biotechnology | Cat# sc-1665, RRID: AB_628069 | WB (1:1000) |
| Antibody | Anti-Rb antibody (mouse Monoclonal) | Santa Cruz Biotechnology | Cat# sc-74562, RRID: AB_2177334) | WB (1:1000) |
| Antibody | Anti-p-Rb antibody (rat monoclonal) | Santa Cruz Biotechnology | Cat# sc-56175, RRID: AB_785453 | WB (1:1000) |
| Antibody | Anti-E2F-1 antibody (mouse monoclonal) | Santa Cruz Biotechnology | Cat# sc-137059, RRID: AB_2096771 | WB (1:1000) |
| Antibody | Anti-E2F-2 antibody (rabbit polyclonal) | Santa Cruz Biotechnology | Cat# sc-633, RRID: AB_2096793 | WB (1:1000) |
| Antibody | Anti-NFκB p65 antibody (mouse monoclonal) | Santa Cruz Biotechnology | Cat# sc-71675, RRID: AB_1126640 | WB (1:1000) |
| Peptide, recombinant protein | IL-1β human | Sigma Aldrich | SRP6169 | 10 ng/mL |
| Commercial assay or kit | Diacetyl dichlorofluorescein staining | Sigma Aldrich | 35848 | |
| Commercial assay or kit | EdU Flow Cytometry Assay Kits | Invitrogen | C10425 | |
| Commercial assay or kit | CCK-8 assay | KeyGen | KGA317s-3000 | |
| Commercial assay or kit | Propidium iodide staining | KeyGen | KGA512 | |
| Commercial assay or kit | IL-1β, IL-6 and TNF-α ELISA kit | KeyGen | KGEMC001b-1; KGEMC004-1; KGEMC102a-1. | |
| Commercial assay or kit | Vectastain Elite ABC reagent | Fisher Scientific | NC9461324 | |
| Commercial assay or kit | Protein Extraction Kit | Thermo Fisher | AM1556 | |
| Commercial assay or kit | Lipofectamine2000 | Thermo Fisher | 11668019 | |
| Commercial assay or kit | ECL | Beyotime | P0018FS | |
| Commercial assay or kit | TRIzol reagent | Beyotime | R0016 | |
| Commercial assay or kit | PrimeScript RT Master Mix | TaKaRa | Cat. #RR036Q | |
| Commercial assay or kit | ChIP kit | Cell Signaling Technology | #9005 | |
| Chemical compound, drug | Rapamycin | Sigma Aldrich | R8781 | 50 nM |
| Software, algorithm | SPSS | SPSS | RRID: SCR_002865 | |
| Software, algorithm | GraphPad | GraphPad Prism | RRID: SCR_002798 | |

## Human NP collection

Thirty-two fresh human intervertebral disc tissue samples were harvested from patients undergoing intervertebral disc surgery at the First Affiliated Hospital of Nanjing Medical University (patients' information is listed in *Supplementary file 2*). Before the operation, the informed consents of the patients were obtained. These consents included the voluntary donation of the diseased nucleus pulposus tissue extracted from the operations, and consent for the use of all specimens for scientific research and for publication of the results obtained in scientific journals. This project was implemented by the approval of the Ethics Committee of the First Affiliated Hospital of Nanjing Medical University (registered number 2018 SR-233). All the samples were divided into four groups using the Pfirrmann score, which was determined on the basis of MRI results from each patient before surgery.

## Human NP cell isolation and culture

The NP tissues were cut into small pieces and digested at 37°C overnight with collagenase XI (Sigma, Ohio, USA), dispase II (Sigma, Ohio, USA) and cell culture medium (containing 2% penicillin/ streptomycin and 10% fetal bovine serum, Thermo, Massachusetts, USA). The cell solution was centrifuged to obtain the cell pellets. NP cells were stimulated with 10 ng/mL IL-1β (Sigma, Ohio, USA) to establish a degeneration model (*Shen et al., 2017*). We also cotreated IL-1β-treated NP cells with 50 nM rapamycin (*Gao et al., 2018*) (Sigma, Ohio, USA) to determine its function during NP cell degeneration.

## Plasmid transfection and siRNA interference

Human p16 plasmid vectors and siRNA were provided by Invitrogen (Massachusetts, USA) and GenePharma (Shanghai, China). The *CDKN2A* gene was inserted into the pcDNA3.1 plasmid. On the basis of the manufacturer's instructions, NP cells were transfected with Lipo6000 (Beyotime, Shanghai, China). Then cells were transfected by siRNA against *CDKN2A* (*Lau et al., 2007*). 24 hr later, IL-1β (10 ng/mL) was used to treat the cells for 4 days, and the cells were harvested for subsequent experiments. The transfection efficiency was examined by quantitative real-time polymerase chain reaction (qRT-PCR), immunofluorescence (IF), and western blot (WB).

## p16 KO mice and tail suspension (TS) IVDD model

The *Cdkn2a* heterozygous mice (FVB N2 background) were a gift from Baojie Li (Shanghai Jiao Tong University, Shanghai, China) and had been backcrossed on the C57BL/6J background. These mice were mated to produce *Cdkn2a* knock-out (p16 KO) and wild-type (WT) littermates. Animal use was approved by the Institutional Animal Care and Use Committee of Nanjing Medical University (approval number: IACUC-1709021). As an IVDD model (*Hutton et al., 2002*; *Nakamura et al., 2013*), TS was carried out in mice for 4 weeks. A specialized cage was made to suspend the mice (*Figure 4—figure supplement 1A*). Forty-eight 16-week-old WT and p16 KO mice were randomly separated into two groups that were divided into two subgroups: WT control (WT), p16 KO control (p16 KO), tail-suspended WT (WT+TS), and tail-suspended p16 KO (p16 KO+TS) groups. At the appropriate time point, the mice were humanely killed, and the lumbar vertebrae (from lumbar 1 to 5) were removed for examination.

## Immunofluorescence

Cultured NP cells were rinsed three times using phosphate-buffered saline (PBS), fixed by 4% formaldehyde for 15 min, incubated in 0.25% Triton X-100 for 15 min, and blocked by 5% bovine serum albumin (BSA, Sigma, Ohio, USA) in PBS for 30 min at room temperature. Then, the cells were treated with primary antibody against p16 (ab51243, Abcam, Cambridge, UK) for one night at 4°C, and incubated with DyLight 488-conjugated goat anti-rabbit IgG antibody (Abbkine, California, USA) for 2 hr in the dark at room temperature. The cells were visualized using a fluorescence microscope (Leica, Wetzlar, Germany), and nuclei were counterstained with DAPI (Beyotime, Shanghai, China).

## Radiological study

Before the animals were sacrificed, X-ray images and micro-MRI scans were taken. X-rays were used to measure the disc height and vertebral body. The intervertebral disc height index (DHI) was

obtained by calculating the average values of the posterior, middle and anterior parts of the inter-vertebral disc, and these values were divided by the average height of the adjacent vertebral body (*Figure 4—figure supplement 4*). Micro-MRI was performed using T2-weighted sections. On the basis of changes in signal intensity with four grades (1, normal; 2, minimal decrease; 3, moderate decrease; and 4, severe decrease), a revised Thompson classification was used to evaluate disc status.

## Histological staining

Mouse lumbar spines (L3–L6) were decalcified for 14 days after fixation in 4% paraformaldehyde (PFA) solution. Human NP tissues and mouse spines were processed for paraffin embedding and sectioning into 5-μm-thick slices for histological staining or immunohistochemistry (IHC), as described below. To evaluate disc degeneration, deparaffinization, hydration, and hematoxylin and eosin (H and E) staining were used to treat the paraffinized slices (*Wang et al., 2009*) so that cells and tissue morphology could be observed. Masson's stain (*Nagatoya et al., 2002*) was added to analyze NP fibrosis; Safranin O (*Kiviranta et al., 1985*) was added to analyze proteoglycans (PGs); and senescence-associated beta-galactosidase (SA-β-gal) (*Ji et al., 2012*) was added to identify senescent cells.

## IHC

IHC was performed as in a previous study (*Yukata et al., 2018*). Briefly, sections were treated with sodium citrate (10 mM, 100°C) (for antigen retrieval) and $H_2O_2$ (10% in PBS) (for endogenous peroxidase inactivation). Next, the slices were blocked with 10% goat serum, and incubated overnight at 4°C using primary antibodies against β-galactosidase (ab203749, Abcam, Cambridge, UK), 8-hydroxy-2 deoxyguanosine (8-OHdG) (ab48508, Abcam, Cambridge, UK), Ki67 (ab15580, Abcam, Cambridge, UK), and PCNA (ab92552, Abcam, Cambridge, UK). Then, biotinylated goat anti-mouse or anti-rabbit IgG (Sigma, Ohio, USA) were used to treat the slices, before they were incubated with Vectastain Elite ABC reagent (Fisher Scientific, Hampton, New Hampshire, USA) for 30 min. 3,3-diaminobenzidine was used for staining, followed by counterstaining with hematoxylin. 8-OHdG, Ki67, and PCNA are mostly expressed in the nucleus; p16 and β-galactosidase are expressed in both the nucleus and the cytoplasm. The positive cell rate for 8-OHdG, Ki67 and PCNA is the ratio of the number of positive nuclei to the number of all hematoxylin-labeled cells. The positive cell rate for p16 and β-galactosidase is the ratio of the number of positive nuclei or/and cytoplasm to the number of all hematoxylin-labeled cells.

## Protein extraction and western blot (WB)

Proteins were harvested from human NP tissues or mouse disc tissues with a Protein Extraction Kit (Thermo, Massachusetts, USA). Immunoblotting was performed as in a previous study (*Miao et al., 2008*), using primary antibodies, against collagen I/X (ab34710/ab58632, Abcam, Cambridge, UK), collagen II (ab34712, Abcam, Cambridge, UK), Sirt1 (ab110304, Abcam, Cambridge, UK), superoxide dismutase 1/2 (SOD1/2) (ab13498/ab13533, Abcam, Cambridge, UK), matrix metalloproteinases-13 (MMP-13) (ab52915/ab39012, Abcam, Cambridge, UK), nuclear factor kappa-B-p65 (NF-κB-p65) (SC-71675, Santa Cruz, California, USA), insulin-like growth factor 1 (IGF-1) (ab9572, Abcam, Cambridge, UK), vascular endothelial growth factor (VEGF) (ab69479, Abcam, Cambridge, UK), p19/53 (SC-1665/SC-126, Santa Cruz, California, USA), cyclin-dependent kinases 4/6 (CDK4/6) (ab199728/ab131469, Abcam, Cambridge, UK), retinoblastoma protein/phosphorylated retinoblastoma protein (Rb/pRB) (SC-74562/SC-56175, Santa Cruz, California, USA), transcription factor E2F1/2 (E2F1/2) (SC-137059/SC-633, Santa Cruz, California, USA), and β-actin (ab8226, Abcam, Cambridge, UK). Immunoreactive bands were analyzed by Scion Image Beta 4.02 and visualized with ECL (Beyotime, Shanghai, China).

## RNA extraction and quantitative real-time PCR (qRT-PCR)

Total RNA was harvested from human NP cells and mouse disc tissues with TRIzol reagent (Beyotime, Shanghai, China). PrimeScript RT Master Mix (Perfect Real Time, TaKaRa, California, USA) was used to reverse transcribe RNA to cDNA. *Supplementary file 3* tabulates the qRT-PCR primer

sequences. GAPDH was used for normalization. Relative mRNA expression levels were determined by the $2^{-\Delta\Delta Ct}$ method.

## Enzyme-linked immunosorbent assay (ELISA)

Serum samples were obtained from blood collected from the eyeballs of mice in each group. Mouse NP cells were also collected by the method described above. The levels of IL-1β, IL-6 and TNF-α in NP cell supernatants were determined using an ELISA kit (KeyGen, Nanjing, China).

## Flow cytometry analyses

Total ROS production, NP cell proliferation and cell-cycle progression were separately assessed using diacetyl dichlorofluorescein staining (Sigma Aldrich, Ohio, USA), propidium iodide staining (KeyGen, Nanjing, China), and EdU Flow Cytometry Assay Kits (Invitrogen, Massachusetts, USA), respectively. Human and mouse NP single-cell suspensions were prepared in PBS, and the cells were treated with the corresponding specialized reagent. The cell pellets were incubated at 37°C for 30 min and obtained by centrifugation. Finally, the specimens were investigated by flow cytometry with a FACSCalibur flow cytometer.

## CCK-8 cell viability assay

Cell proliferation was evaluated with a CCK-8 assay (KeyGen, Nanjing, China). In brief, cells in each group (5000/well) were allowed to grow for 24, 48, and 72 hr. Ten microliters of CCK reagent in a total volume of 100 µl was put into each well, before incubation for 3–4 hr. The absorbance at 450 nm was measured by an ELISA plate reader (Thermo Electron, Massachusetts, USA).

## Chromatin immunoprecipitation (ChIP) assay

The 2000-bp region upstream of the p16 gene was selected as the promoter region according to the National Center for Biotechnology Information database (http://www.ncbi.nlm.nih.gov/). After predicting DNA-binding sites for the NF-κB-p65 transcription factor in the p16 promoter using the JASPAR core database (*Bryne et al., 2008*), five putative binding sites were identified close to the transcription start site. ChIP primers targeting these sites were designed by Primer Premier. ChIP assays were carried out using a ChIP kit (CST, Massachusetts, USA) and a p65 antibody obtained from Abcam. The relative binding of NF-κB-p65 to p16 was assessed by PCR, followed by digital imaging of agarose gels.

## Plasmid constructs and luciferase reporter gene assay

WT and mutant *CDKN2A* gene promoter segments were synthesized by Promoterbio Lab (Taizhou, China) and then cloned into the pGL4.23-basic luciferase vector to obtain the pGL4.23-p16-wt and pGL4.23-p16-mut plasmids: control plasmid (0.1 µg)+pGL4.23-basic vector (0.1 µg)+Renilla plasmid (0.01 µg), control plasmid+pGL4.23-WT p16 promoter vector+Renilla plasmid, control plasmid +pGL4.23-mutant p16 promoter vector+Renilla plasmid, NF-κB-p65 sequence plasmid+pGL4.23-ba-asic vector+Renilla plasmid, NF-κB-p65 sequence plasmid+pGL4.23-WT p16 promoter vector +Renilla plasmid, and NF-κB-p65 sequence plasmid+pGL4.23-mutant p16 promoter vector+Renilla plasmid, separately with Lipofectamine2000 (Thermo Fisher, Massachusetts, USA). The cells were incubated in normal culture medium for 48 hr after transfection. Luciferase assays was implemented after the cells were collected and lysed. And then luciferase activity was standardized to Renilla luciferase activity.

## Statistical analyses

All analyses were carried out by SPSS software (version 20.0, USA). Mean ± SD was used to present the data. To compare differences between groups, one-way ANOVA and student's t-test were used. After analysis using a chi-square test, qualitative data are presented as percentages. All graphs were generated using GraphPad Software (version 5.0.0, USA). P values were two-sided, and $p < 0.05$ indicated statistical significance.

## Acknowledgements

This work was supported by Australia National Health and Medical Research Council (NHMRC) (1158402, 1127396), National Natural Science Foundation of China (NSFC) project grant (81572149, 81671928, 81730066), and State Scholarship Fund by the China Scholarship Council (CSC201908080215).

## Additional information

### Funding

| Funder | Grant reference number | Author |
|---|---|---|
| National Health and Medical Research Council | 1158402 | Liping Wang |
| National Natural Science Foundation of China | 81572149 | Yongxin Ren |
| China Scholarship Council | CSC201908080215 | Hui Che |
| National Natural Science Foundation of China | 81671928 | Cory J Xian Liping Wang |
| National Health and Medical Research Council | 1127396 | Cory J Xian |
| National Natural Science Foundation of China | 81730066 | Dengshun Miao |

The funders had no role in study design, data collection and interpretation, or the decision to submit the work for publication.

### Author contributions

Hui Che, Jie Li, Conceptualization, Investigation, Methodology; You Li, Huan Liu, Jingyi Qin, Zhen Zhang, Data curation, Formal analysis, Investigation; Cheng Ma, Data curation, Investigation; Jianghui Dong, Methodology; Cory J Xian, Investigation, Writing - original draft, Writing - review and editing; Dengshun Miao, Conceptualization; Liping Wang, Conceptualization, Resources, Supervision, Funding acquisition, Project administration; Yongxin Ren, Conceptualization, Resources, Supervision, Funding acquisition

### Author ORCIDs

Hui Che http://orcid.org/0000-0002-3345-2033
Jianghui Dong https://orcid.org/0000-0003-3961-1688
Cory J Xian http://orcid.org/0000-0002-8467-2845
Liping Wang https://orcid.org/0000-0001-9355-1167

### Ethics

Human subjects: This work was implemented by the approval of the Ethics Committee of The First Affiliated Hospital of Nanjing Medical University (approval number: 2018-SR-233). Before the operation, the informed consents of the patients have been obtained, including the patient's voluntary donation of the diseased nucleus pulposus tissue extracted from the operation, and their consents that all specimens will be used for scientific research and the results obtained will be published in scientific journals.

Animal experimentation: Animal use was approved by the Institutional Animal Care and Use Committee of Nanjing Medical University (approval number: IACUC-1709021).

### Decision letter and Author response

Decision letter https://doi.org/10.7554/eLife.52570.sa1
Author response https://doi.org/10.7554/eLife.52570.sa2

## Additional files

### Supplementary files

- Supplementary file 1. Primer sequences for CHIP.
- Supplementary file 2. Patients' information.
- Supplementary file 3. Primer sequences for RT-PCR.
- Transparent reporting form

### Data availability

All data generated or analysed during this study are included in the manuscript and supporting files.

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
