## [Decision Letter]

**Acceptance summary:**

The reviewers and editors found the work highly significant in enhancing our current understanding of the mechanism(s) of intervertebral disc degeneration, which is the hallmark of the crippling aging disorder, osteoarthritis. The model was considered adequate to establish a role of p16 in regulating the cell cycle and oxidative stress. The review concerns were also thoughtfully and thoroughly addressed, which added to the high level of enthusiasm.

**Decision letter after peer review:**

Thank you for submitting your article "p16 deficiency protects against intervertebral disc degeneration" for consideration by *eLife*. Your article has been reviewed by two peer reviewers, and the evaluation has been overseen by a Reviewing Editor and Clifford Rosen as the Senior Editor. The following individual involved in review of your submission has agreed to reveal their identity: Chao Xie (Reviewer #2).

Below is the summary of our discussion with the reviewers to help you prepare a revised submission..

Summary:

Both reviewers felt that the submission merited further consideration due to its overall novelty with regard to the role of P16 deficiency in human intervertebral disc degeneration, and the approach used for the study. Specifically, the study design, methods, and material were seen to be well described and the data collections presented clearly. However, there was unanimity among reviewers regarding the lack of critical information in certain areas, as well as the need for validations and further details, all of which are specified below.

Essential revisions:

1) There are issues with the Abstract and Introduction. The Abstract did not fully cover the findings in the manuscript. For example, the in vitro work in Figures 1-3 were not reflected in the Abstract. In addition, the significance of this work and the potential impact was not mentioned. Likewise, the Introduction did not provide enough information for the readers. For example, in the second paragraph of the Introduction, although the authors listed the characteristics of the microenvironment, it is still unclear why the microenvironment would make discs degenerate more easily. Another example is in the third paragraph of the Introduction, wherein it is unclear which tissue was examined.

2) The quality of the Figure 1A is poor. It was hard for the reviewer to see the multinuclear giant cells at this magnification from HE staining. Please provide counterstaining for P16. Also, please comment on p16 expression in the matrix in G5. Also, in Figure 1B, the criteria of p16 positive cells are not explained. Thus, positive cells in G2 may look different from those in G5 – please provide more detail.

3) Patient details are lacking. It is suggested that, in the Materials and methods section, a table should provide patient details, including age, sex and Pfirrmann score.

4) It remains unclear from the data generated from the human specimens whether the MRI Pffirmann Grade correlated individually with p16 expression. In other words, please provide the p16 expression level of each individual (IHC) and their corresponding MRI Pffirmann Grade as a supplement for Figure 1. Please also provide high power MRI images of mice lumbar intervertebral and MRI Pffirmann Grade.

5) Instead of randomly choosing regions of interest (ROI) for H&E, Safranin O, Masson, and p16 IHC staining (Figure 1), please use adjacent slices for all of these labels, which should help document a relationship between different molecules, as has been done in Figure 5A.

6) Certain issues regarding mechanical loading of the spine through tail suspension used for the weightlessness simulation study need clarification. Please explain how this model could represent aging-related human IVDD? What is the validity and fidelity of this model as it relates to lumbar intervertebral disc degeneration?

7) Please clarify any differences in ventral vs. dorsal portions of the nucleus pulposus and vertebral endplate in the mechanical loading study.

---

## [Author Response]

Essential revisions:1) There are issues with the Abstract and Introduction. The Abstract did not fully cover the findings in the manuscript. For example, the in vitro work in Figures 1-3 were not reflected in the Abstract. In addition, the significance of this work and the potential impact was not mentioned. Likewise, the Introduction did not provide enough information for the readers. For example, in the second paragraph of the Introduction, although the authors listed the characteristics of the microenvironment, it is still unclear why the microenvironment would make discs degenerate more easily. Another example is in the third paragraph of the Introduction, wherein it is unclear which tissue was examined.

We appreciated your comments. We have improved the Abstract section with more information about the findings and the potential impact of this work. In Introduction, we have added details about how the microenvironment would make discs degenerate more easily and the tissue types associated with aging-associated disorders.

2) The quality of the Figure 1A is poor. It was hard for the reviewer to see the multinuclear giant cells at this magnification from HE staining. Please provide counterstaining for P16. Also, please comment on p16 expression in the matrix in G5. Also, in Figure 1B, the criteria of p16 positive cells are not explained. Thus, positive cells in G2 may look different from those in G5 – please provide more detail.

We appreciated your comments. We have improved the quality of images in Figure 1A. In the Materials and methods, we have added details about criteria of p16 positivity as well as the other immuno-positive cells. p16 is mainly presented in the cells. Although the NP in G5 is severely degenerated, we did not think p16 is significantly observed in the matrix.

3) Patient details are lacking. It is suggested that, in the Materials and methods section, a table should provide patient details, including age, sex and Pfirrmann score.

We appreciated your comments. We have added the patients’ information including age, sex and Pfirrmann score in the Materials and methods section.

4) It remains unclear from the data generated from the human specimens whether the MRI Pffirmann Grade correlated individually with p16 expression. In other words, please provide the p16 expression level of each individual (IHC) and their corresponding MRI Pffirmann Grade as a supplement for Figure 1. Please also provide high power MRI images of mice lumbar intervertebral and MRI Pffirmann Grade.

We appreciated your comments. We have added the p16-positive cell ratio with its corresponding MRI Pffirmann Grade as Figure 1—figure supplement 2. We have magnified the MRI images of mouse lumbar intervertebra as Figure 1—figure supplement 1 and added the Pffirmann Grade as Figure 1—figure supplement 2.

5) Instead of randomly choosing regions of interest (ROI) for H&E, Safranin O, Masson, and p16 IHC staining (Figure 1), please use adjacent slices for all of these labels, which should help document a relationship between different molecules, as has been done in Figure 5A.

We appreciated your comments. We have updated the H&E, Safranin O, Masson, and p16 IHC staining in Figure 1.

6) Certain issues regarding mechanical loading of the spine through tail suspension used for the weightlessness simulation study need clarification. Please explain how this model could represent aging-related human IVDD? What is the validity and fidelity of this model as it relates to lumbar intervertebral disc degeneration?

We really appreciated your comments and suggestions. We have clarified more details about the feasibility of tail suspension as a model related to IVDD in the Discussion section. The basic role of the disc is a mechanical function such as load distribution, energy dissipation, and motion permission Mechanical factor is proposed as one of necessary mechanisms for accelerating the IVDD of both human and rodent disc via altered loading. Weightlessness simulation by TS changes hydrostatic pressure in the disc and leads to the ECM destruction, inflammatory response and a catabolic process representing premature aging-related IVDD progression (Foldes et al., 1996). Therefore, we believe TS mouse model is an applicable mechanical representation of tensile force on human disc as well as the aging process.

7) Please clarify any differences in ventral vs. dorsal portions of the nucleus pulposus and vertebral endplate in the mechanical loading study.

We really appreciated your comments. Due to the physiological curvature of the spine, the ventral portions suffer more critical pressure compared with the dorsal portions. Umehara et al. reported that the posterolateral sections of the disc containing NP and vertebral endplate bear lower elastic modulus and presents a milder degenerated degree compared to other portions (Umehara S, et al: Effects of degeneration on the elastic modulus distribution in the lumbar intervertebral disc. Spine. 1996; 1; 21(7):811-9; discussion 820). Ebara et al. stated that the posterolateral regions of the disc sustained lower values of stress compared with the anterior regions in a tensile properties study (Ebara S, et al. Tensile properties of nondegenerate human lumbar anulus fibrosis. Spine. 1996; 15; 21(4):452-61). In addition, it has been found that the ability to withstand applied stress was significantly reduced in the middle portion of the disc (Fujita Y et al. Radial tensile properties of the lumbar annulus fibrosus are site and degeneration dependent. J Orthop Res. 1997; 15(6):814-9)